# Estimation of Individual Positive Anti-Islet Autoantibodies from 3 Screen ICA Titer

**DOI:** 10.3390/ijms25147618

**Published:** 2024-07-11

**Authors:** Eiji Kawasaki, Hideaki Jinnouchi, Yasutaka Maeda, Akira Okada, Koichi Kawai

**Affiliations:** 1Diabetes, Thyroid, and Endocrine Center, Shin-Koga Hospital, Kurume 830-8577, Japan; 2Department of Internal Medicine, Jinnouchi Hospital Diabetes Care Center, Kumamoto 862-0976, Japan; hideaki@jinnouchi.or.jp; 3Minami Diabetes Clinical Research Center, Clinic Masae Minami, Fukuoka 815-0071, Japan; myas555@minami-cl.jp; 4Okada Clinic, Fukuoka 812-0053, Japan; okada@okadaclinic.or.jp; 5Kawai Clinic, Ibaraki 305-0812, Japan; info@kawai-clinic.com

**Keywords:** type 1 diabetes, GAD autoantibodies, IA-2 autoantibodies, ZnT8 autoantibodies, 3 Screen ICA, bridging-type ELISA

## Abstract

The 3 Screen ICA ELISA is a novel assay capable of simultaneously measuring autoantibodies to glutamic acid decarboxylase (GADA), insulinoma-associated antigen-2 (IA-2A), and zinc transporter 8 (ZnT8A), making it a valuable tool for screening type 1 diabetes. Despite its advantages, it cannot specify which individual autoantibodies are positive or negative. This study aimed to estimate individual positive autoantibodies based on the 3 Screen ICA titer. Six hundred seventeen patients with type 1 diabetes, simultaneously measured for 3 Screen ICA and three individual autoantibodies, were divided into five groups based on their 3 Screen ICA titer. The sensitivities and contribution rates of the individual autoantibodies were then examined. The study had a cross-sectional design. Sixty-nine percent (424 of 617) of patients with type 1 diabetes had 3 Screen ICA titers exceeding the 99th percentile cut-off level (20 index). The prevalence of GADA ranged from 80% to 100% in patients with a 3 Screen ICA over 30 index and 97% of patients with a 3 Screen ICA ≥300 index. Furthermore, the prevalence of all individual autoantibodies being positive was 0% for ≤80 index and as high as 92% for ≥300 index. Significant associations were observed in specific titer groups: the 20–29.9 index group when all the individual autoantibodies were negative, the 30–79.9 index group when positive for GADA alone or IA-2A alone, the 30–299.9 index group when positive for ZnT8A alone, the 80–299.9 index group when positive for both IA-2A and ZnT8A, the 300–499.9 index group when positive for both GADA and ZnT8A, and the ≥300 index group when positive for all individual autoantibodies. These results suggest that the 3 Screen ICA titer may be helpful in estimating individual positive autoantibodies.

## 1. Introduction

Type 1 diabetes is characterized by the selective destruction of pancreatic β-cells, resulting in an absolute deficiency of endogenous insulin secretion. This condition arises from a combination of genetic predisposition and environmental factors that trigger an immune response against pancreatic β-cells [1,2,3,4,5,6,7,8]. Etiologically, type 1 diabetes is broadly divided into “immune-mediated” and “idiopathic” types [9,10]. The involvement of autoimmunity in “immune-mediated” type 1 diabetes is evidenced by the presence of circulating anti-islet autoantibodies against β-cell autoantigens, which are valuable markers for predicting and diagnosing type 1 diabetes [11,12,13,14,15,16,17]. Recent reports, including our own, highlight the clinical utility of the 3 Screen ICA ELISA, which can simultaneously measure three major anti-islet autoantibodies, glutamic acid decarboxylase autoantibodies (GADA), insulinoma-associated antigen-2 autoantibodies (IA-2A), and zinc transporter 8 autoantibodies (ZnT8A) [18,19,20,21,22,23,24]. The use of 3 Screen ICA ELISA requires only 1/4 the amount of serum compared to individual autoantibody assays, and it can also be used with capillary samples [20,25]. It has also been reported that the sensitivity is 10–20% higher than that of individual autoantibodies with similar specificity in newly diagnosed type 1 diabetes [26]. Thus, this simple, sensitive, and specific assay is a convenient and cost-effective one-step strategy for screening patients with immune-mediated type 1 diabetes in clinical practice, and therefore it has clinical usefulness compared with individual autoantibody assays. Although this assay is a valuable screening tool for immune-mediated type 1 diabetes, a positive result does not specify which individual autoantibodies are present or absent.

It has also been reported that both the specificities of particular autoantibodies and the number of positive anti-islet autoantibodies are crucial for predicting the onset and progression of slowly progressive type 1 diabetes (SPIDDM) [27,28,29]. Therefore, this study aimed to estimate individual positive autoantibodies based on the 3 Screen ICA titer.

## 2. Results

### 2.1. Prevalence of Individual Autoantibodies in 3 Screen ICA Titer Groups

As shown in Figure 1, the distribution of 3 Screen ICA titers in autoantibody-positive patients was similar between acute-onset type 1 diabetes and SPIDDM. Therefore, this study was conducted in all patients without dividing them into subtypes of type 1 diabetes. Patients were classified into five groups based on the 10th, 25th, 75th, and 90th percentiles of the 3 Screen ICA titer among autoantibody-positive cases: 20–29.9 index; 30–79.9 index; 80–299.9 index; 300–499.9 index; and ≥500 index. Figure 2 illustrates the prevalence of individual autoantibodies in each group. The prevalence of GADA ranged from 80 to 100% in all groups with a 3 Screen ICA titer of ≥30 index. Furthermore, 97% of patients with an index of ≥300 were positive for GADA, while only 3% were single or double positive for IA-2A and ZnT8A.

### 2.2. Associations between 3 Screen ICA Titer and the Combination of Individual Autoantibodies

As shown in Table 1, all patients who were negative for three individual autoantibodies belonged to the 20–29.9 index group. The frequency of patients positive for GADA alone was highest in the 300–499.9 index group (30%), with 24% and 26% of GADA-single positive patients belonging to the 30–79.9 index and 80–299.9 index groups, respectively. Patients positive for IA-2A alone were most common in the 30–79.9 index group (38%), followed by the 80–299.9 index group (33%). The highest percentage of patients positive for ZnT8A alone was in the 30–79.9 index group and the 80–299.9 index group (38%). For the combination of two individual autoantibodies, GADA/IA-2A-positive patients had the highest prevalence in the 300–499.9 index group (38%), GADA/ZnT8A-positive patients in the 300–499.9 index group (55%), and IA-2A/ZnT8A-positive patients in the 80–299.9 index group (62%). The frequency of patients who were positive for all three individual autoantibodies was highest in the 300–499.9 index group (58%), followed by the ≥500 index group (35%), and the prevalence of all individual autoantibodies being positive was 0% for ≤80 index, and as high as 92% for ≥300 index.

We performed a chi-squared test on these results using contingency table analysis and calculated the post hoc cell contribution rate. As a result, the contribution rate in all individual autoantibody-negative patients was 16.16 in the 20–29.9 index group, which exceeded 1.96 (*p* < 0.05). Similar statistical analysis showed significant associations in the 30–79.9 index group when positive for GADA alone or IA-2A alone, the 20–79.9 index group when positive for ZnT8A alone, the 80–299.9 index group when positive for both IA-2A and ZnT8A, the 300–499.9 index group when positive for both GADA and ZnT8A, and the ≥300 index groups when positive for all individual autoantibodies.

### 2.3. Associations between 3 Screen ICA Titer and the Number of Individual Autoantibodies

Table 2 shows the relationship between 3 Screen ICA titers and the number of individual anti-islet autoantibody positives. The frequency of patients who were positive for zero individual autoantibodies was 100% in the 20–29.9 index group. Patients positive for 1 autoantibody were most common in the 80–299.9 index group and the 300–499.9 index group (27%), followed by the 30–79.9 index group (26%). The frequency of patients positive for 2 autoantibodies was highest in the 300–499.9 index group (44%). The frequency of patients who were positive for all three individual autoantibodies was highest in the 300–499.9 index group (58%), followed by the ≥500 index group (35%). Similarly, patients with two or more autoantibodies were also most frequent in the 300–499.9 index group (49%), followed by the ≥500 index group (24%). The post hoc cell contribution ratio exceeded 1.96 in the 20–29.9 index group for the zero autoantibody-positive patients, the 30–79.9 index group for one autoantibody-positive patients, the 300–499.9 index group for two autoantibody-positive patients, and the ≥300 index groups for three autoantibody-positive patients and ≥2 autoantibody-positive patients.

### 2.4. Comparison of 3 Screen ICA Titers among the Individual Autoantibody Combination Groups

Figure 3 illustrates the box-and-whisker plots of 3 Screen ICA titers for each autoantibody group. The lowest median titer of 3 Screen ICAs was observed in the patients negative for all individual autoantibodies (23.9 index, range 20.1–28.2), followed by those positive for ZnT8A alone (61.7 index, range 20.8–279.8), IA-2A alone (75.3 index, range 21.4–468.8), or IA-2A/ZnT8A (178.7 index, range 63.6–488.8). Notably, patients who were positive for GADA exhibited higher 3 Screen ICA titers even when they were GADA single positive. The median 3 Screen ICA titer was 279.0 index (range 22.5–609.1) for GADA alone, 426.2 index (range 44.4–606.0) for GADA/ZnT8A, and 438.2 index (range 71.8–607.0) for GADA/IA-2A. The highest median titer of 3 Screen ICA was observed in the all-positive group (467.6 index, range 97.8–611.5). The Bonferroni multiple comparison test showed statistically significant differences in 3 Screen ICA titers for all two-group comparisons (corrected *p* < 0.005).

### 2.5. Factors Associated with 3 Screen ICA-Positive but Negative for Individual Autoantibodies

As shown in Table 1, 19 patients were positive for 3 Screen ICA but negative for all individual autoantibodies in the 20–29.9 index group. To analyze the factors associated with this discrepant result, we performed a univariate and multivariate logistic regression analysis using 204 patients who were negative for all individual autoantibodies (Table 3). Univariate logistic regression analysis showed that blood glucose level at the time of serum sampling (Odds ratio 1.01, 95%CI 1.00–1.01, *p* = 0.013) and GADA titer (odds ratio 5.60, 95%CI 3.00–10.5, *p* < 0.0001) were associated with 3 Screen ICA positivity. However, multivariate logistic regression analysis revealed GADA titer as the sole independent factor associated with 3 Screen ICA-positive/individual autoantibody-negative status (odds ratio 7.63, 95%CI 3.28–17.7, *p* < 0.0001).

Furthermore, 3 Screen ICA titers significantly correlated with the GADA titer (Figure 4; R = 0.80, *p* < 0.0001). 

The optimal cut-off value of the GADA for distinguishing between 3 Screen ICA-positive and -negative patients was determined using ROC curve analysis, which identified a value of 2.2 U/mL with a sensitivity of 95% and a specificity of 90% (Appendix A).

## 3. Discussion

In this study, we aimed to estimate individual positive autoantibodies based on the 3 Screen ICA titer. Additionally, we investigated the possible reasons some patients with low 3 Screen ICA titers were negative for all three individual autoantibodies despite both assays using the same principles of bridging-type ELISA. Previously, we reported that the 3 Screen ICA ELISA is a valuable screening tool for patients with immune-mediated type 1 diabetes, potentially increasing diagnostic sensitivity and accuracy beyond existing GADA, IA-2A, and ZnT8A tests [18]. 

The 3 Screen ICA ELISA measures the combined levels of three anti-islet autoantibodies, making it difficult to identify individual positive autoantibodies. However, it has been reported that the autoantibody specificities and the number of positive anti-islet autoantibodies are crucial for predicting the onset and progression of type 1 diabetes, especially in the case of SPIDDM [27,28,29]. In the present study, post hoc cell contribution rates were calculated after chi-squared tests, revealing that each autoantibody combination was significantly associated with specific 3 Screen ICA index groups. Furthermore, the relationship between the number of positive individual autoantibodies and the specific 3 Screen ICA index group was demonstrated. These results suggest that the 3 Screen ICA titer may be helpful in estimating the specificities and number of positive individual autoantibodies. Indeed, there was an association between the median titers of 3 Screen ICA and individual autoantibody groups (Figure 3). Median 3 Screen ICA titers were lower in patients without GADA and differed between groups, even among GADA-positive patients. However, it is important to note that 20–30% of patients with a ≥30 index were positive for GADA alone, regardless of their 3 Screen ICA titer. Excluding patients who were positive for GADA alone, 97.2% of patients with a ≥300 index were positive for two or more autoantibodies, suggesting it might be possible to predict the onset of type 1 diabetes and the progression of SPIDDM based on the 3 Screen ICA titer. Future prospective studies using 3 Screen ICA will be necessary to validate this hypothesis. Moreover, GAD also localizes in tissues other than the islet, whereas IA-2 and ZnT8 are specific to the islet [30,31,32]. This difference in autoantibody titers is likely due to the immune response to GAD being driven by maturation and expansion related to repeated immune stimulation. In contrast, the immune response to IA-2 and ZnT8 results from rapid maturation against immunogenic epitopes during β-cell destruction.

Opinions are divided on whether 3 Screen ICA should be used as a screening test or diagnostic tool for type 1 diabetes. When used as a screening test, it is necessary to measure individual autoantibodies to diagnose type 1 diabetes. Following the Japanese health insurance policy, GADA is always the first anti-islet autoantibody to be measured, and allowances to measure IA-2A are only made when GADA is negative. The positivity and titer of GADA have been reported to be useful in predicting the progression of SPIDDM [28,33], but such sequential measurements are costly and time-consuming. In contrast, the 3 Screen ICA ELISA, which can simultaneously measure multiple anti-islet autoantibodies, is advantageous for rapidly diagnosing type 1 diabetes when its specificity is sufficient. In this study, 64% of cases in the 20–29.9 index group were negative for all individual autoantibodies, even when the 3 Screen ICA was positive. 

Is the 3 Screen ICA showing a false positive in these patients? Considering the results of multivariate logistic regression analysis, the correlation between 3 Screen ICA and GADA titers, as well as the optimal cut-off of GADA (2.2 U/mL), it is presumed that the serum of patients who are negative for all individual autoantibodies but positive for 3 Screen ICA may contain trace amounts of GADA that cannot be detected by the GADA ELISA method. Although both the GADA ELISA and the 3 Screen ICA ELISA kits use the same assay principle, there is a possibility that the sensitivity of the GADA test differs between the two assays because the 3 Screen ICA ELISA can detect trace amounts of GADA below the cut-off value of the GADA ELISA kit. Therefore, 3 Screen ICA positivity and all individual autoantibody negativity can be considered true positives for anti-islet autoantibodies. When 3 Screen ICA is positive, type 1 diabetes can be diagnosed without measuring individual autoantibodies.

This study has some limitations. First, most samples were collected from long-standing type 1 diabetes patients. Since GADA is known to be more persistently positive than other anti-islet autoantibodies [34], the duration of the disease may affect the proportions of individual autoantibodies. Therefore, further investigations involving a larger cohort of patients with newly diagnosed type 1 diabetes are needed. Second, in this study, patients with acute-onset type 1 diabetes and SPIDDM were combined to ensure the number of cases in each 3 Screen ICA titer group. Although the distribution of titers in 3 Screen ICA-positive individuals was similar between both subtypes, it will be necessary to investigate each subtype in the future to confirm our findings. This assay also has some limitations. It was affected by hemolysis in the low range, which results a slight decrease in specificity [20]. Furthermore, islet antigens immobilized on the ELISA plate are more likely to be saturated than individual autoantibody kits, resulting in lower than actual autoantibody titers in the high range [18]. In order to determine the superiority of this kit, it is necessary to prove in a prospective study that the 3 Screen ICA titer is useful for preclinical staging and predicting the onset of type 1 diabetes, without measuring individual autoantibodies, in the first-degree relatives with type 1 diabetes and the general population.

In conclusion, our study suggests that the 3 Screen ICA ELISA can be used as a diagnostic tool for type 1 diabetes without measuring GADA, IA-2A, or ZnT8A. Additionally, the 3 Screen ICA titer may be useful in estimating individual positive autoantibodies.

## 4. Materials and Methods

### 4.1. Patients

Serum samples were collected from 617 Japanese patients with type 1 diabetes, including 436 with acute-onset type 1 diabetes and 181 with SPIDDM. Inclusion criteria included patients with acute-onset type 1 diabetes or SPIDDM who have a history of outpatient or hospitalization at five primary healthcare clinics specialized in diabetes. There were no restrictions on age, sex, or duration of disease. Exclusion criteria included unknown type of diabetes or type 2 diabetes. Patients’ clinical data were extracted from electronic medical records at each participating hospital and gathered by manually filling out a case questionnaire created for this study. A diagnosis of type 1 diabetes was made according to the criteria of the Japan Diabetes Society [35,36]. SPIDDM was diagnosed if patients tested positive for anti-islet autoantibodies at any point during the disease course, regardless of their status at the time of the study. The concept of SPIDDM is different from latent-autoimmune diabetes in adults (LADA) in terms of age of onset and other clinical features [36]. Among the 617 patients, 98 of 612 (16.0%) had complications with other autoimmune diseases (AID), including autoimmune thyroid diseases, at the time of serum sampling, and the remaining 5 patients had an unknown status. The patients used in this study were a subset of those used in the previous study, and their clinical and immunological profiles are outlined in Appendix A. Blood samples were drawn from peripheral veins, either fasting or postprandial, into the collection tube containing clot activator and separator gel at any time, and sera were obtained after centrifugation and stored at −20 °C until autoantibody measurement. Furthermore, it was confirmed that there were no signs of hemolysis which may affect the antibody titer of 3 Screen ICA in all samples.

### 4.2. Autoantibody Assays

The 3 Screen ICA ELISA kit (RSR Ltd., Cardiff, UK) is based on the bridging-type principle with full-length GAD65, the intracellular domain of IA-2 (aa604–979), and dimeric carboxy-terminal domains of ZnT8 (aa275–369) carrying either 325Trp or 325Arg [18]. As the 3 Screen ICA ELISA is a bivalent assay, autoantibodies in serum link to both the antigen on the ELISA plate and the biotinylated antigen, thereby potentially increasing the specificity. Autoantibody levels were expressed as an index defined as (OD of the test sample/ OD of the reference preparation) × 100. A reference preparation was included in every assay. The cut-off value was 20.0 index based on the 99th percentile of 159 healthy control subjects [18]. The inter-assay and intra-assay coefficient variation (CV) using GADA-single-positive, IA-2A-single-positive, and ZnT8A-single-positive sera were as follows: inter-assay CV values of 2.2%, 3.0%, and 3.8% (n = 7) and intra-assay CV values of 3.4%, 1.5%, and 4.0% (n = 5), respectively. In the Islet Autoantibody Standardization Program (IASP) 2020 workshop (Lab ID 1801), the assay sensitivities and specificities achieved were 96.0% and 100%, respectively.

Individual autoantibodies (GADA, IA-2A, and ZnT8A) were also measured in the same sera as the 3 Screen ICA using bridging-type ELISA kits (RSR Ltd.) with corresponding biotinylated proteins, following previously described methods [18]. The results were read from a calibration curve constructed in the same run as the calibrators and expressed in U/mL. The cut-off values were 5.0 U/mL for GADA, 0.6 U/mL for the IA-2A, and 10 U/mL for the ZnT8A. In the IASP 2020 workshop, the assay sensitivities and specificities achieved for GADA, IA-2A, and ZnT8A were 90.0% and 97.8%, 72.0% and 97.8%, and 76.0% and 98.9%, respectively.

### 4.3. Statistical Analysis

Continuous data are expressed as mean ± standard deviation (SD) or median (range). Comparative categorical data analyses were performed using the chi-squared test for trend tests and post hoc cell contribution rate to compare more than two groups. The post hoc cell contribution rate test is a standardized residual test that provides significant information among groups when the absolute value exceeds 1.96. Differences in non-parametric data were evaluated using the Mann–Whitney *U* test or Kruskal–Wallis test, followed by Bonferroni’s multiple comparison test. Correlations between autoantibody titers were analyzed using Spearman’s rank correlation test. Univariate and multivariate logistic regression analyses were performed to test for the association of 3 Screen ICA-positive/individual autoantibody-negative status with variables such as subtype of type 1 diabetes, age at diagnosis, duration of diabetes, gender, body mass index, the levels of hemoglobin A1c, blood glucose, and C-peptide at the time of serum sampling, cooccurrence of other autoimmune diseases, GADA titer, IA-2A titer, and ZnT8A titer. The receiver operating characteristic (ROC) curve analysis determined the optimal cut-off value. A *p* value <0.05 was considered statistically significant. Statistical analysis was performed using StatView statistical software (version 5.0; SAS Institute, Cary, NC, USA) and SigmaPlot software (version 14.0, Systat Software Inc., San Jose, CA, USA).

## Figures and Tables

**Figure 1 ijms-25-07618-f001:**
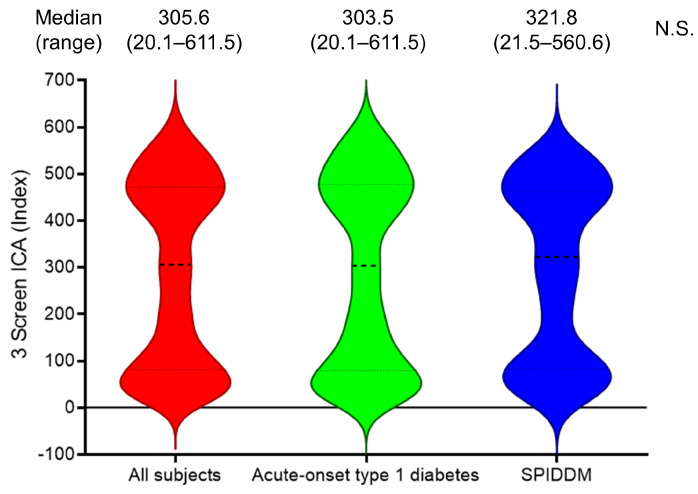
Distribution of 3 Screen ICA index in subjects with type 1 diabetes. Dashed lines indicate median (central), first (lower), and third quartile (upper) titers. There was no significant difference in the median titer of 3 Screen ICA between acute-onset type 1 diabetes and SPIDDM. N.S., not significant.

**Figure 2 ijms-25-07618-f002:**
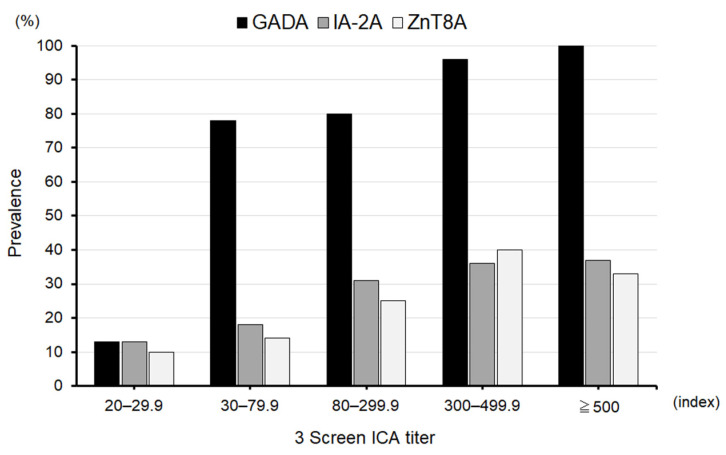
Prevalence of GADA, IA-2A, and ZnT8A in each 3 Screen ICA titer group.

**Figure 3 ijms-25-07618-f003:**
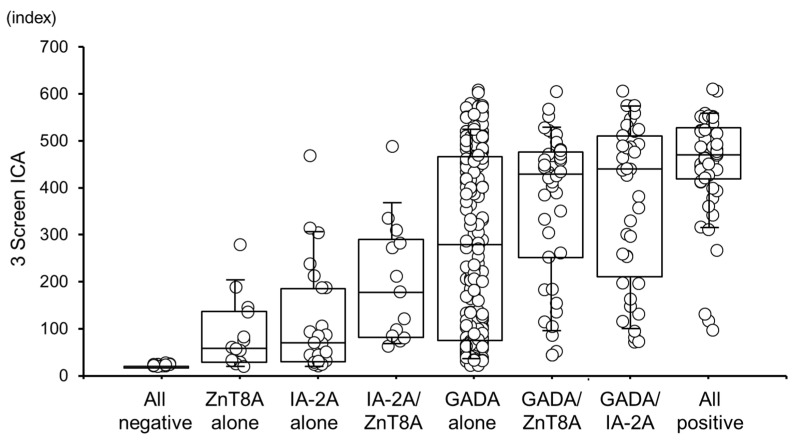
Box-and-whisker diagram of 3 Screen ICA titer in each individual autoantibody group. Horizontal bars indicate median levels. Bonferroni’s multiple comparison tests showed statistically significant differences in 3 Screen ICA titers for all two-group comparisons (corrected *p* < 0.005).

**Figure 4 ijms-25-07618-f004:**
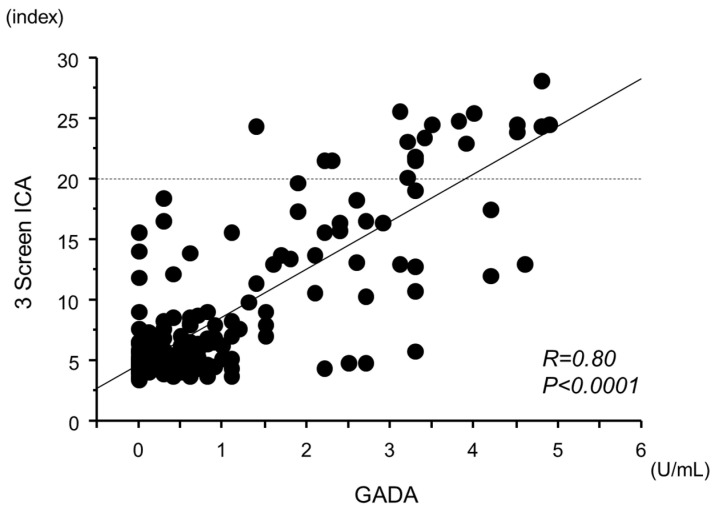
Correlation between the levels of 3 Screen ICA and GADA in individual autoantibody-negative patients. A significant positive correlation was observed between 3 Screen ICA titers and GADA titers (R = 0.80, *p* < 0.0001).

**Table 1 ijms-25-07618-t001:** Relationship between 3 Screen ICA titer and positive individual anti-islet autoantibodies.

		3 Screen ICA Titer (Index)				
Group	n	20–29.9 (n = 30)	30–79.9 (n = 74)	80–299.9 (n = 104)	300–499.9 (n = 141)	≥500 (n = 75)
All negative	19	**19 (100)**	0 (0)	0 (0)	0 (0)	0 (0)
GADA alone	222	4 (2)	**53 (24)**	58 (26)	67 (30)	40 (18)
IA-2A alone	24	4 (17)	**9 (38)**	8 (33)	3 (13)	0 (0)
ZnT8A alone	13	**3 (23)**	**5 (38)**	5 (38)	0 (0)	0 (0)
GADA/IA-2A	39	0 (0)	2 (5)	12 (31)	15 (38)	10 (26)
GADA/ZnT8A	42	0 (0)	3 (7)	9 (21)	**23 (55)**	7 (17)
IA-2A/ZnT8A	13	0 (0)	2 (15)	**8 (62)**	3 (23)	0 (0)
All positive	52	0 (0)	0 (0)	4 (8)	**30 (58)**	**18 (35)**

Bold letters indicate significant post hoc cell contribution rate (>1.96).

**Table 2 ijms-25-07618-t002:** Relationship between 3 Screen ICA titer and number of individual anti-islet autoantibody positives.

		3 Screen ICA Titer (Index)				
Group	n	20–29.9 (n = 30)	30–79.9 (n = 74)	80–299.9 (n = 104)	300–499.9 (n = 141)	≥500 (n = 75)
0 autoantibodies (+)	19	**19 (100)**	0 (0)	0 (0)	0 (0)	0 (0)
1 autoantibody (+)	259	11 (4)	**67 (26)**	71 (27)	70 (27)	40 (15)
2 autoantibodies (+)	94	0 (0)	7 (7)	29 (31)	**41 (44)**	17 (18)
3 autoantibodies (+)	52	0 (0)	0 (0)	4 (8)	**30 (58)**	**18 (35)**
≥2 autoantibodies (+)	146	0 (0)	7 (5)	33 (23)	**71 (49)**	**35 (24)**

Bold letters indicate significant post hoc cell contribution rate (>1.96).

**Table 3 ijms-25-07618-t003:** Identification of factors associated with 3 Screen ICA positivity in individual autoantibody-negative patients by univariate and multivariate logistic regression analysis.

Factors	Univariate Analysis		Multivariate Analysis	
	OR (95%CI)	*p* Value	OR (95%CI)	*p* Value
Acute-onset type	0.94 (0.32–2.75)	N.S.		
Male	1.68 (0.61–4.62)	N.S.		
Age at diagnosis	1.00 (0.98–1.03)	N.S.		
Duration	0.97 (0.93–1.01)	N.S.		
BMI	0.91 (0.80–1.05)	N.S.		
HbA1c	1.21 (0.80–1.82)	N.S.		
Blood glucose	1.01 (1.00–1.01)	0.013	1.00 (0.99–1.01)	N.S.
C-peptide	0.89 (0.50–1.59)	N.S.		
Autoimmune disease (+)	0.48 (0.06–3.80)	N.S.		
GADA titer	5.60 (3.00–10.5)	<0.0001	7.63 (3.28–17.7)	<0.0001
IA-2A titer	0.62 (0.0–788.1)	N.S.		
ZnT8A titer	1.33 (0.94–1.90)	N.S.		

N.S., not significant.

## Data Availability

The datasets generated and/or analyzed during the current study are available from the corresponding author upon reasonable request.

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
