# Peer review of "Estimation of Individual Positive Anti-Islet Autoantibodies from 3 Screen ICA Titer"

_ijms, 2024, doi:10.3390/ijms25147618_

Round 1
Reviewer 1 Report
Comments and Suggestions for Authors
This manuscript was aimed to estimate individual positive autoantibodies based on the
3 Screen ICA titter, novel assay capable of simultaneously measuring GADA, IA-2A, and ZnT8A. They included 617 patients with T1D, simultaneously measured for 3 Screen ICA and three individual autoantibodies, and allocated them into 5 groups based on their 3 Screen ICA. The authors concluded that the 3 Screen ICA ELISA can be used as a diagnostic tool for T1D without measuring GADA, IA-2A, or ZnT8A. Additionally, the 3 Screen ICA titer may be useful in estimating individual positive autoantibodies.The article has clinical importance, keeping in mind the necessity of implementing new assays for easier detection of autoantibodies to pancreatic antigens, due to request for diagnosis or detection of risk for T1D.
Comments to the Authors:
1. Please put the study design in the Abstract section
2. Please explain more clearer 20 index in Abstract section, lines 20, 21
3. In Methodology section please add inclusion and exclusion criteria for investigated groups
4. Please explain SPIDDM, is it LADA group or not, because LADA by definition include only patients age 35 or older, and in the Table 1 I can see patients with age 8?
5. Line 234…are the samples taken early in the morning, after fasting or not, venous blood? which tube, please add
6. Please, add information about the institutions and medical care level where recruitment took place…
7. Please add more literature data, 11 references are not enough
Author Response
We thank you for your evaluation of our manuscript. We have revised the manuscript according to your comments. In the revised manuscript, all changes made in the manuscript are marked using “Track changes”.
Comments and Suggestions for Authors
- Please put the study design in the Abstract section
Response: According to your comment, we added a sentence “The study had a cross-sectional design.” in the Abstract section.
- Please explain more clearer 20 index in Abstract section, lines 20, 21
Response: As the 20 index is a 99 percent cut-off level, we added this information in the Abstract section.
- In Methodology section please add inclusion and exclusion criteria for investigated groups
Response: Thank you for your comment. We added the following sentences in the Materials and Methods section.
“Inclusion criteria included acute-onset type 1 diabetes or SPIDDM who have a history of outpatient or hospitalization at five primary health care clinics specialized in diabetes. There were no restrictions on age, sex, or duration of disease. Exclusion criteria included unknown type of diabetes or type 2 diabetes.”
- Please explain SPIDDM, is it LADA group or not, because LADA by definition include only patients age 35 or older, and in the Table 1 I can see patients with age 8?
Response: SPIDDM and LADA are not the same. In addition to adult-onset cases, SPIDDM also includes cases that develop in childhood. Most of these children with SPIDDM are patients whose diabetes was diagnosed with urinary glucose screening tests at schools (Urakami T et al. Diabetes Res Clin Pract 80, 473-476, 2008). We added the sentences “The concept of SPIDDM is different from latent-autoimmune diabetes in adults (LADA) in terms of age of onset and other clinical features [18].” in the Materials and Methods section.
- Line 234…are the samples taken early in the morning, after fasting or not, venous blood? which tube, please add
Response: We added the sentence “Blood samples were drawn from peripheral veins, either fasting or postprandial, into the collection tube containing clot activator and separator gel at any time, and sera were obtained after centrifugation and stored at -20℃ until autoantibody measurement.” in the Materials and Methods section.
- Please, add information about the institutions and medical care level where recruitment took place…
Response: All participating hospitals belong primary health care clinics specialized in diabetes. We added this information in the Materials and Methods section.
- Please add more literature data, 11 references are not enough.
Response: According to your comment, we added 8 references regarding this subject in the revised manuscript.

Reviewer 2 Report
Comments and Suggestions for Authors
The manuscript entitled Estimation of Individual Positive Anti-Islet Autoantibodies from 3 Screen ICA Titer is an original article. The authors estimate individual positive autoantibodies based on the 3 Screen ICA titer. The 3 Screen ICA ELISA is a novel assay capable of simultaneously measuring autoantibodies to GAD, IA-2, and ZnT8, making it a valuable tool for diagnosing type 1 diabetes.
They concluded that the 3 Screen ICA titer may be helpful in estimating individual positive autoantibodies.
The manuscript is well written. However, there is a major issue of this manuscript.
Major revision
From a total of 11 references (which are very few), 7 are about the same group of researchers. Therefore, only 4 references constitute the literature synthesis. How do you comment this? There are no others data regarding this subject?
Is this study a slice of pie type article? Do you use in this article data from an already published study? If it is the case, then the authors must include a comment in study limitations.
The first part of the conclusions are reproduced in introduction as already known (voir lines 43-47). Obviously, in this case, it is hard to think that this study bring something new. Please clarify.
Author Response
We thank you for your evaluation of our manuscript. We have revised the manuscript according to your comments. In the revised manuscript, all changes made in the manuscript are marked using “Track changes”.
Comments and Suggestions for Authors
The manuscript is well written. However, there is a major issue of this manuscript.
- From a total of 11 references (which are very few), 7 are about the same group of researchers. Therefore, only 4 references constitute the literature synthesis. How do you comment this? There are no others data regarding this subject?
Response: According to your comment, we added 8 references regarding this subject in the revised manuscript.
- Is this study a slice of pie type article? Do you use in this article data from an already published study? If it is the case, then the authors must include a comment in study limitations.
Response: This paper is not a slice of pie type article, but rather an extension study using a subset of the subjects used in our previous study (Kawasaki E et al. J Diabetes Investig 2023, 1081-1091). We added the phrase “The patients used in this study were a subset of those used in the previous study,” in the Materials and Methods section.
- The first part of the conclusions are reproduced in introduction as already known (voir lines 43-47). Obviously, in this case, it is hard to think that this study bring something new. Please clarify.
Response: Thank you for your comment. We have previously reported that 3 Screen ICA ELISA may be a valuable screening tool for Japanese patients with type 1 diabetes (Kawasaki E et al. J Diabetes Investig 2023, 1081-1091).
Therefore, we have reworded the sentence in the Introduction section to be more accurate: "Although this assay is a valuable screening tool for immune-mediated type 1 diabetes, a positive result does not specify which individual autoantibodies are present or absent.".

Reviewer 3 Report
Comments and Suggestions for Authors
The recent study presented the use of the 3 Screen ICA ELISA for the diagnosis of type 1 diabetes (T1D). This seems to be both intriguing and practical. Nevertheless, please address the following issues.
1, The clinical usefulness of this 3 Screen ICA ELISA has to be introduced in detail, highlighting its merits.
2. How can data be gathered from 617 Japanese individuals with T1D? Kindly provide a comprehensive and thorough description.
3. The Islet Autoantibody Standardization is crucial in this test. How can one adhere to the specified criteria?
4. No product comparison allowed. How can one determine the superiority of this kit?
5. Clear elucidation of the limits of this assay must be done, notwithstanding the disclosure of limitations in the present report.
6. The 3 Screen ICA ELISA is a valuable tool for assessing specific autoantibodies in individuals, without the need to measure GADA, IA-2A, or ZnT8A. Nevertheless, the information presented in the current paper does not support this notion.
Author Response
We thank you for your evaluation of our manuscript. We have revised the manuscript according to your comments. In the revised manuscript, all changes made in the manuscript are marked using “Track changes”.
Comments and Suggestions for Authors
The recent study presented the use of the 3 Screen ICA ELISA for the diagnosis of type 1 diabetes (T1D). This seems to be both intriguing and practical. Nevertheless, please address the following issues.
- The clinical usefulness of this 3 Screen ICA ELISA has to be introduced in detail, highlighting its merits.
Response: According to your comment, we added the clinical usefulness of this 3 Screen ICA ELISA in detail in the Introduction section.
- How can data be gathered from 617 Japanese individuals with T1D? Kindly provide a comprehensive and thorough description.
Response: Patients’ clinical data were extracted from electronic medical records at each participating hospital and gathered by manually filling out a case questionnaire created for this study. We added this information in the Materials and Method section.
- The Islet Autoantibody Standardization is crucial in this test. How can one adhere to the specified criteria?
Response: The titers of anti-islet autoantibodies in this test are calculated relative to the reference preparation included in the kit, and therefore rely on the selection of cut-off value by the user to adhere to the specified criteria for the islet autoantibody standardization. Furthermore, it is recommended that each laboratory include its own panel of control samples in the assay. Using a low cut-off will maximize the detection of samples that would be positive in 3 Screen ICA assays but a high proportion of these are likely to be positive for one individual autoantibody only. A higher cut-off would give greater assay specificity and will detect a greater proportion of samples that are positive for two or more individual autoantibodies. Therefore, the selection of a cut-off value for 3 Screen ICA by the user should depend on the screening objective and the type of population being screened.
- No product comparison allowed. How can one determine the superiority of this kit?
Response: Thank you for your comment. In order to determine the superiority of this kit, we believe that it is necessary to prove in a prospective study that the 3 Screen ICA titer is useful for preclinical staging and predicting the onset of type 1 diabetes, without measuring individual autoantibodies, in the first-degree relatives with type 1 diabetes and the general population. We added this thought in the Discussion section.
- Clear elucidation of the limits of this assay must be done, notwithstanding the disclosure of limitations in the present report.
Response: As the limitations of this assay, we added the following sentences in the Discussion section.
“This assay also has some limitations. It was affected by hemolysis in the low range which results a slight decrease in specificity [8]. Furthermore, islet antigens immobilized on the ELISA plate are more likely to be saturated than individual autoantibody kits, resulting in lower than actual autoantibody titers in the high range [6].” In addition, we added the sentence in the Materials and Methods section describing that all samples were confirmed that there was no sign of hemolysis.
- The 3 Screen ICA ELISA is a valuable tool for assessing specific autoantibodies in individuals, without the need to measure GADA, IA-2A, or ZnT8A. Nevertheless, the information presented in the current paper does not support this notion.
Response: As you pointed out, the notion that the 3 Screen ICA ELISA is a valuable tool for assessing specific autoantibodies in individuals, without the need to measure GADA, IA-2A, or ZnT8A is not presented in this study. As described in the last sentences in the Discussion section, we demonstrated that type 1 diabetes can be diagnosed without measuring individual autoantibodies when 3 Screen ICA is positive, and that the 3 Screen ICA titer may be useful in estimating individual positive autoantibodies.

Round 2
Reviewer 1 Report
Comments and Suggestions for Authors
The authors corrected the manuscript according to the suggestions, I have no further requirements.
Author Response
We thank you for your favorable re-evaluation.
Reviewer 2 Report
Comments and Suggestions for Authors
Thank you for responding to my comments.
Author Response

(The authors gave the same response as above.)

Reviewer 3 Report
Comments and Suggestions for Authors
It has been revised according to comments.
Author Response

(The authors gave the same response as above.)
